

# Acute ischemic preconditioning does not influence high-intensity intermittent exercise performance

Isabela Coelho Marocolo[1], Gustavo Ribeiro da Mota[1], André Monteiro Londe[1], Stephen D. Patterson[2], Octávio Barbosa Neto[1] and Moacir Marocolo[3]

[1] Human Performance and Sport Research Group, Department of Sport Sciences, Institute of Health Sciences, Federal University of Triangulo Mineiro, Uberaba, MG, Brazil
[2] School of Sport, Health, and Applied Science, St. Mary's University, Twickenham, UK
[3] Physiology and Human Performance Research Group, Department of Physiology, Institute of Biological Sciences, Federal University of Juiz de Fora, Juiz de Fora, MG, Brazil

## ABSTRACT

This study evaluated the acute effect of ischemic preconditioning (IPC) on a high-intensity intermittent exercise performance and physiological indicators in amateur soccer players. Thirteen players (21.5 ± 2 yrs) attended three trials separated by 3–5 days in a counterbalanced randomized cross-over design: IPC (4 × 5-min occlusion 220 mmHg/reperfusion 0 mmHg) in each thigh; SHAM (similar to the IPC protocol but "occlusion" at 20 mmHg) and control (seated during the same time of IPC). After 6-min of each trial (IPC, SHAM or control), the players performed the YoYo Intermittent Endurance Test level 2 (YoYoIE2). The distance covered in the YoYoIE2 (IPC 867 ± 205 m; SHAM 873 ± 212 m; control 921 ± 206 m) was not different among trials ($p = 0.10$), furthermore, lactate concentration and rate of perceived exertion did not differ ($P > 0.05$) among protocols. There were also no significant differences in either mean heart rate (HR) or peak HR ($p > 0.05$) for both IPC and SHAM compared to control. Therefore, we conclude that acute IPC does not influence high-intensity intermittent exercise performance in amateur soccer players and that rate of perceived exertion, heart rate and lactate do not differ between the intervention IPC, SHAM and control.

Corresponding author
Gustavo Ribeiro da Mota,
grmotta@gmail.com

## INTRODUCTION

The technique of ischemic preconditioning (IPC) alternates brief periods of occlusion and re-establishment (e.g., 3–5 min each) of muscle blood flow prior to exercise using a tourniquet applied in the proximal part of a limb (e.g., lower limbs) (*Marocolo et al., 2016a*) aiming to improve exercise performance. The ergogenic effects are still debatable (*Marocolo et al., 2016a*) especially because some study designs are limited (e.g., without placebo-control) (*Da Mota & Marocolo, 2016*) and the mechanism behind IPC as well as its supposed exercise enhancement has yet to be clarified. Potential mechanisms of IPC exercise enhancement are hyperemia (5–6 fold increase in muscle blood flow) during reperfusion (*Libonati et al., 2001*), attenuated ATP depletion, and increased

phosphocreatine production/oxygen uptake during the reperfusion phase (*Andreas et al., 2011*). Additionally, IPC has been suggested to improve aerobic and anaerobic performance (*Cruz et al., 2015*; *Incognito, Burr & Millar, 2016*) mostly in high-intensity exercise in cyclic modalities (e.g., running, cycling, swimming).

Although one study (*Gibson et al., 2015*) evaluated the IPC effect in athletes with a history of participation in team sport and others in repeated sprint cycling (*Patterson et al., 2015*), no study has examined its effects on high-intensity intermittent exercise (i.e., specific to team sports). The YoYo intermittent endurance test level 2 (YoYoIE2) simulates high-intensity intermittent exercise similar to the activity seen in soccer, and is effective due to its simplicity and reproducibility (*Bradley et al., 2011*), and it is correlated with both oxygen uptake and citrate synthase activity in untrained men (*Krustrup et al., 2015*). Although *Krustrup et al. (2015)* stated that aerobic energy production plays a "more important role" for fatigue resistance in untrained subjects versus trained male soccer players during the YoYoIE2, their results showed also an important contribution of the glycolytic pathway in the YoYoIE2 (e.g., blood lactate $\sim$11 mmol at 5-min of recovery). Since IPC seems to improve performance in exercise tests of predominantly lactic anaerobic and aerobic capacity (*Incognito, Burr & Millar, 2016*), the aim of this study was to evaluate the acute effect of IPC on YoYoIE2 performance and some physiological parameters in amateur soccer players. We hypothesized that IPC would increase the distance covered in YoYoIE2, because this test for our specific sample (amateur players) would be suitable to recruit predominantly both the glycolytic and aerobic systems and, therefore, meet the ergogenic potential from IPC (*Cruz et al., 2015*; *Incognito, Burr & Millar, 2016*).

## MATERIALS AND METHODS

Thirteen male amateur soccer players ($21.5 \pm 2$ yrs, $173 \pm 5$ cm, $69.7 \pm 6$ kg, $12.6 \pm 5$ body fat %) volunteered for this study. The players participated in $2.7 \pm 0.5$ h wk$^{-1}$ of training and $1.3 \pm 0.2$ matches per week and playing experience of $11 \pm 4$ yrs. Inclusion criteria were: (a) soccer playing experience more than 5 yrs, (b) no smoking history during the last year, (c) absence of any cardiovascular or metabolic disease, (d) systemic blood pressure lower than 140/90 mmHg and no use of antihypertensive medication, (e) no use of creatine supplementation, anabolic steroids, drugs or medication with potential effects on physical performance (self-reported) and (g) no recent musculoskeletal injury. This study was approved by the local institutional Ethical Committee for Human Experiments (Federal University of Triangulo Mineiro—993.636/2015) and was performed in accordance with ethical standards in sports science research (*Harriss & Atkinson, 2015*). In addition, all subjects signed an informed consent form. Based on prior research (*Bradley et al., 2011*), a sample size less than 13 (i.e., *n* between six and nine) was sufficient to detect a significant ($p < 0.05$) difference among playing positions ("group") in the YoYoIE2 (main dependent variable). Also, a study using a similar IPC protocol (independent variable) and physical test duration, reported significant statistical effects with a sample size of 12 individuals (*Cruz et al., 2015*). Thus, to counteract any potential drop out, a sample of 13 subjects was included for this study.

## Experimental design of the study

Subjects attended the laboratory five times (with 3–5 days in-between) (*Marocolo et al., 2016a*), for initial screening and anthropometric measurements and for familiarization with the equipment and proceedings. In the 3rd, 4th and 5th visits, a randomized crossover assignment (IPC, SHAM and control) was adopted and the YoYoIE2 was carried out after 6 min of each trial (IPC, SHAM or control). All YoYoIE2 tests were conducted by the same experienced researcher in a constant environment (23 ± 2 °C; Humidity: 75 ± 4%) at the same time of the day (9:00–11:00 h). To prevent the possibility of a placebo (positive) effect (*Marocolo et al., 2017*; *Marocolo et al., 2016b*; *Marocolo et al., 2016c*), all subjects were informed that all conditions could improve performance. Also, in order to prevent nocebo (negative) effects the subjects were informed that IPC and SHAM would cause absolutely no harm, despite discomfort related to the maneuvers (*Ferreira et al., 2016*). Additionally, the tester was blinded for which protocol (i.e., IPC, SHAM or control) the subjects had undergone before. Also, the subjects were kept blinded in relation to performance and other indicators until the end of the research, i.e., no information about distance covered (the audio of speed and level of YoYoIE2 was in unknown language), HR and lactate. Coffee (or caffeine products), tea, and alcohol intake was prohibited as well as strenuous exercise for 48 h before testing (*Incognito, Burr & Millar, 2016*).

## Ischemic preconditioning protocol

The IPC maneuver consisted of four cycles of 5-min occlusion (220 mmHg) and 5-min reperfusion (no pressure) in each thigh (total duration 40 min), using a pneumatic tourniquet (Riester Komprimeter; Riester, Jungingen, Germany) administered at the subinguinal region of the thighs. We applied this specific IPC protocol for the following reasons: (a) several studies have successfully explored the ergogenic effects of IPC on exercise performance with the same protocol (*Cruz et al., 2015*; *Jean-St-Michel et al., 2011*; *Marocolo et al., 2016a*; *Patterson et al., 2015*); (b) at least three ischemia-reperfusion cycles are necessary to protect against skeletal muscle infarction and endothelial dysfunction after prolonged periods of imposed ischemia (*Pang et al., 1995*); and (c) it is considered safe and well tolerated in both patients and healthy volunteers (*Gonzalez et al., 2014*). The occlusion and reperfusion phases were conducted alternately between the thighs, with subjects remaining seated (knee at 90° angle). The effectiveness of occlusion in the IPC session was controlled by auscultation of the arteries around the ankle (*Loenneke et al., 2012*). In the SHAM protocol, an external pressure of 20 mmHg was administered, as proposed in previous studies (*Foster et al., 2011*; *Marocolo et al., 2016a*). For the control condition, the subjects remained seated for a respective period of time. Although a consensus does not exist with regards to the time between the last IPC cycle and the exercise testing, ranging from 5 min to 90 min in most studies with exercise performance (*Marocolo et al., 2016a*), the IPC has been shown to improve exercise performance within 45 min of the final cuff inflation (*Bailey et al., 2012*; *Patterson et al., 2015*). Thus, in the current study, the warm-up to the YoYoIE2 was performed after 6 min of each trial (i.e., IPC, SHAM or control).

## Perceived recovery status

Before each session, all subjects indicated a score on a perceived recovery scale (*Laurent et al., 2011*), (from 0 "very little recovered, feeling extremely tired" to 10 "very well recovered, feeling with great energy"), about their relative physical recovery to make sure the player was in the same recovery condition before the trials. If a player scored 4 or less (somewhat recovered) for his recovery status, he was excluded from that day's session and invited to wait until the next day to perform the session.

### Yo-Yo intermittent endurance test level 2 (YoYoIE2)

After 6 min of recovery from each trial (IPC, SHAM or control) the warm-up for the YoYoIE2 began. The warm-up consisted of performing the first three running bouts of the YoYoIE2 followed by a period of lower-extremity stretching. All players were familiar with the YoYoIE2 because they used this test on regular basis for fitness evaluation purpose. The YoYoIE2 consists of a repeated 2 × 20-m shuttle run at progressively increasing speed stages (initial speed ∼12 km.h-1), guided by specific audio (5-s to recovery in a marked 2.5 × 2 m area behind the finishing line). Cessation of the test was assessed by failure to reach the finish line by the tone on two occasions(*Bradley et al., 2011*). The YoYoIE2 is a reproducible, sensitive tool associated with soccer match performance and can differentiate intermittent exercise performance of several standards and is therefore highly suggested for soccer players (*Bradley et al., 2014*; *Bradley et al., 2012*). For consistency, the subjects performed the YoYoIE2 in groups of three or four (always the same group) and they received similar verbal encouragement during the tests.

### Heart rate (HR), rate of perceived exertion (RPE) and blood lactate concentration

HR was monitored throughout the entire YoYoIE2, by an individual RS800CX heart monitor (Polar Electro®, Kempele, Finland). After the YoYoIE2, the player indicated (individually to prevent influence from other player) a score for his RPE via CR-10 Borg scale in order to determine the subjective intensity of the session. This scale ranges from 0 to 10, where 0 is "nothing at all" and 10 is "very very hard (maximal)" (*Borg & Kaijser, 2006*). Blood samples (25 µL) were collected from the fingertip 4 min post YoYoIE2 using a lancet, and the lactate concentration was measured via a valid (*Fell et al., 1998*) portable lactate analyzer (Accutrend® Plus system, ROCHE, Basel Switzerland).

## Statistical analysis

The Shapiro–Wilk test was applied to verify the normal distribution of the data. For between- protocol analysis, one-way analysis of variance (ANOVA) for repeated measures was conducted, followed by post-hoc Tukey's test or nonparametric ANOVA (Friedman test) followed by a post-hoc Dunn's test was performed. Only for the distances covered in the YoYoIE2 we calculated the effect size (ES; Cohen d) to determine the meaningfulness of the difference (practical relevance), classified as: trivial (<0.2), small (>0.2–0.6), moderate (>0.6–1.2), large (>1.2–2.0) and very large (>2.0) as recommended (*Batterham & Hopkins, 2006*). The significance level was set at 0.05. The software used for data analysis was GraphPad® (Prism 6.0, San Diego, CA, USA).

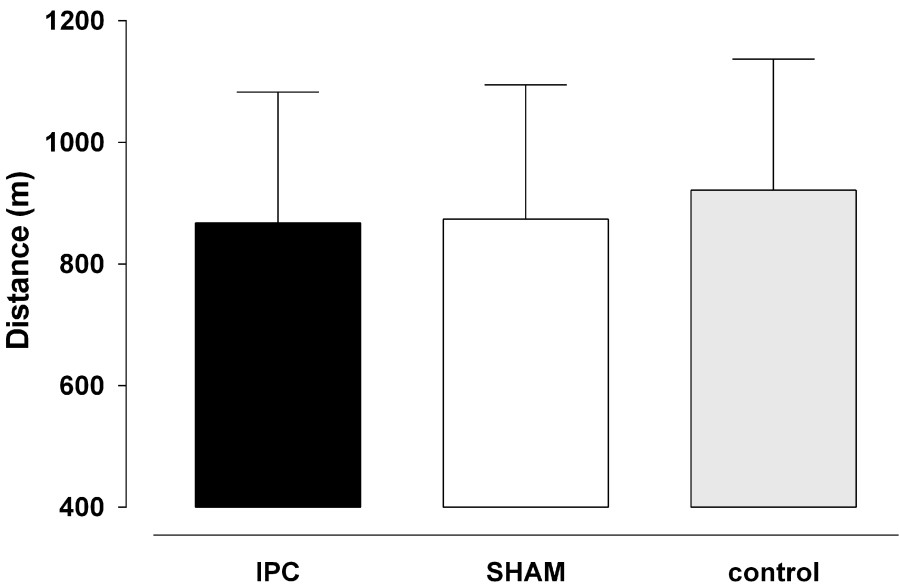

**Figure 1 The distance covered in the YoYoIE2 test for the trials.** IPC, ischemic preconditioning. Values are mean ± standard deviation ($n = 13$).

**Table 1 Rate of perceived exertion, blood lactate concentration, mean and peak heart rate for the trials.** Values are mean standard deviation; ($n = 13$).

| Variable | IPC | SHAM | Control | *p* value |
|---|---|---|---|---|
| Rate of perceived exertion | $9.0 \pm 1.1$ | $8.7 \pm 1.3$ | $8.8 \pm 0.9$ | 0.69 |
| Blood lactate (mmol L$^{-1}$) | $11.1 \pm 2.7$ | $11.4 \pm 3$ | $12.7 \pm 2.4$ | 0.19 |
| HR mean (bpm) | $169 \pm 7$ | $169 \pm 5$ | $171 \pm 6$ | 0.17 |
| HR peak (bpm) | $189 \pm 5$ | $189 \pm 5$ | $191 \pm 4$ | 0.06 |

**Notes.**
Values are mean ±standard deviation; ($n = 13$).
HR, heart rate; IPC, ischemic preconditioning; SHAM, placebo IPC.

## RESULTS

Before the trials, the perceived recovery scores were not different ($P = 0.1$) among the protocols (IPC $= 6.9 \pm 1.7$, SHAM $= 7.7 \pm 1.5$ and control $= 7.7 \pm 0.7$). RPE and lactate concentrations did not differ after the trial among all protocols. Mean HR and peak HR presented no difference ($p = 0.17$ and $p = 0.06$, respectively) among the three trials (Table 1).

The distance covered in the YoYoIE2 did not differ ($P = 0.10$; ES $= 0.16$) among the protocols (Fig. 1). The mean distances were: IPC 867 m ($SD = 205$), SHAM 873 m ($SD = 212$) and control 921 m ($SD = 206$). The effect sizes were 0.26 (small) IPC vs control, 0.2 (trivial) SHAM vs control and 0.03 (trivial) IPC vs SHAM.

Figure 2 shows the individual change in distances covered by each player in the YoYoIE2. It should be noted that only three players (∼23%) performed better after SHAM and four (∼30%) after IPC, while the others virtually covered the same distances or even shorter distances than under control conditions.

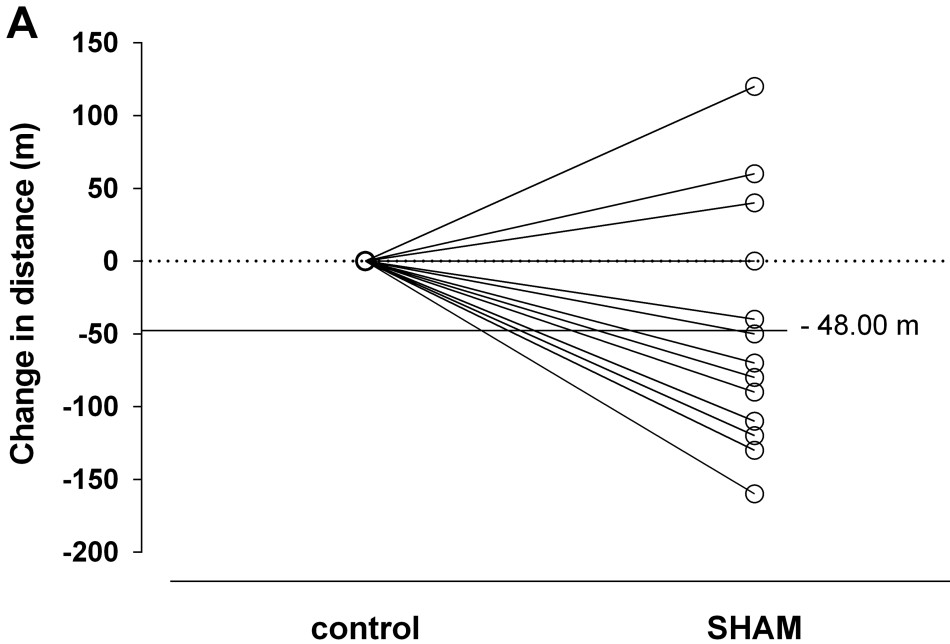

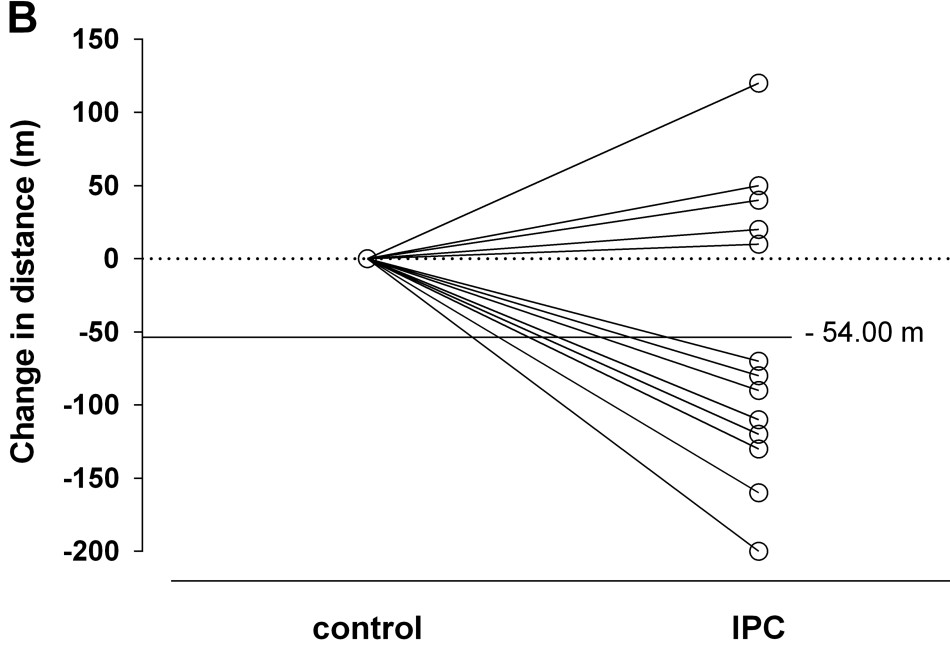

**Figure 2** **Individual changes with SHAM (A) and ischemic preconditioning—IPC (B) on the distance covered in the YoYoIE2 test.** Each black line represents different players ($n = 13$). The horizontal continuous line represents the mean difference of each intervention from control.

## DISCUSSION

The purpose of this study was to evaluate the acute effect of IPC on YoYoIE2 performance and related physiological indicators in amateur soccer players. Our main findings are that acute IPC did not influence performance in the YoYoIE2. In addition, IPC did not influence perceived exertion, blood lactate concentrations (after the test) or either mean or peak HR in comparison with SHAM and control interventions.

Our current study is the first to evaluate the IPC effect on a high-intermittent performance test that involves changing of direction, and acceleration/deceleration (YoYoIE2), all of which is important for team sports, specifically soccer (*Bradley et al., 2011*; *Paul, Gabbett & Nassis, 2016*; *Varley & Aughey, 2013*). The distance covered in the YoYoIE2 did not differ significantly among the three conditions, showing no influence (i.e., neither positive nor negative), as well as no practical relevance (i.e.; trivial effect sizes) of acute IPC on this sort of test and population. The distance covered in the current study in the control conditions ($935 \pm 232$ m) is higher than that reported for untrained men ($665 \pm 271$ m) but lower than highly trained soccer players ($2,027 \pm 298$ m) (*Krustrup et al., 2015*). Since the scores of perception of recovery were similar before all of the three trials (i.e., players starting under same conditions), basic conditions for comparison of different protocols were ensured.

According to a recent systematic review (*Incognito, Burr & Millar, 2016*), the most consistent benefit of IPC would be an improvement in exercise tests with the prevalence of energy coming from glycolytic and aerobic sources. The YoYoIE2 has been designed to stimulate both anaerobic and aerobic pathways; however despite this, IPC had no effect on intermittent exercise lasting $\sim$6 min. These same authors have argued that there are IPC responders (i.e., improvements in performance) and non-responders (i.e., no effects on performance) phenotypes, due to large between-subject variability of results and, therefore, they recommend prudence in the interpretation of mean group changes in exercise performance due to IPC (*Incognito, Burr & Millar, 2016*). In other words, individual analysis should also be performed to more thoroughly assess the potential ergogenic effects of IPC on exercise performance. Analyzing the individual responses of YoYoIE2 in the current study, we found only three and four players better in SHAM and IPC protocols, respectively. These results suggest that acute IPC does not enhance high-intensity intermittent exercise performance for the majority of the players. Considering that the same acute IPC protocol (i.e., $4 \times 5$ cycles of occlusion/reperfusion) improved endurance performance ($\sim$8%) with exercise time duration very close to ours ($\sim$6 min) in recreational cyclists (*Cruz et al., 2015*), we may speculate that the nature of YoYoIE2; i.e., intermittent maximal test, including acceleration and deceleration, changing of directions, important eccentric actions involved, might counterbalance the effect following acute IPC. Future studies should explore this hypothesis more deeply by including other kinds of YoYo test (e.g., recovery level I).

Recent evidence suggests the possibility of a placebo (positive) or nocebo (negative) effect (*Ferreira et al., 2016*; *Marocolo et al., 2015*), but in the current study, this potential bias was controlled by three methods: first, all subjects were informed that both maneuvers

could improve performance (SHAM and IPC) and none of them could harm performance; second, we kept the tester blinded in relation to the previous protocol (i.e., the physical tester did not know if the player had performed IPC, SHAM or control) and third, no information about distance covered, HR and lactate were provided.

The RPE is used in studies as an essential and valid indicator of exercise intensity, in addition to the measurement of traditional physiological variables (*Borg & Kaijser, 2006*). In our study, RPE was checked immediately after each test and reached high values, suggesting near maximal internal intensity in the tests. However, RPE did not differ in any condition. Considering that IPC effects could have any motivational basis on exercise performance (*Marocolo et al., 2015*; *Marocolo et al., 2016b*; *Marocolo et al., 2016c*) the register of the RPE for evaluating psychophysical parameters would assist in intensity measures. Our current results showed that IPC did not affect the RPE and it is consistent with previous studies: after sprints series compared to SHAM (*Gibson et al., 2015*) and after a five-kilometer race to the control intervention (*Bailey et al., 2012*).

Heart rate was measured continuously during the test, and neither peak nor mean HR differed among interventions (IPC, SHAM and CON). One study showed a shortening QT interval during exercise, without any RR interval changes after IPC intervention (*Caru et al., 2016*). Conversely, in another study (*Clevidence, Mowery & Kushnick, 2012*) significantly higher values for the same parameters in cyclists were found after IPC intervention. For these authors, the HR increase could be explained by a decrease in compensation to a reduced stroke volume and therefore cardiac output at the time of the IPC maneuver, as previously described (*Iida et al., 2007*). However, our values of covered distance were not different among interventions; nine and eight subjects covered a little shorter distance after SHAM and IPC, respectively, compared to control, which could explain, at first glance, no difference in values of peak and mean HR during the YoYoIE2 following IPC and SHAM interventions.

In our study, blood lactate concentrations are in line with the literature for YoYoIE2 (*Krustrup et al., 2015*) in similar subjects (i.e., untrained) confirming also the significant anaerobic contribution to ATP supply. Another study showed that IPC generated lower blood lactate concentration at submaximal intensity during an incremental running test (*Bailey et al., 2012*), but in the current study we did not find any difference among IPC, SHAM and control conditions. This difference is probably due to the type of exercise, since in the present study we applied an intermittent maximal test.

There are several limitations to the present study: no comparison among different playing positions (*Da Mota et al., 2016*); the absence of measurements to understand the mechanisms underpinning the IPC/SHAM maneuvers, as well the YoYoIE2, such as muscle blood flow, muscle oxy/deoxyhemoglobin, oxygen uptake, enzyme activities, serum nitric oxide, and partial oxygen/carbon dioxide pressures. Also, considering that reactive oxygen species play an important role in both IPC (*Zhu & Zuo, 2013*) and exercise (*He et al., 2016*), measurements of reactive oxygen species for both IPC and exercise test would be interesting. Our sample size ($n = 13$) was relatively small, and it may generate a type II error. However, it is higher than one study that applied the same IPC protocol and found statistically significant effects from IPC in cyclists performing similar duration of

maximal exercise (*Cruz et al., 2015*). Lastly, despite our control to avoid potential placebo and nocebo effects, we recognize the limited ability to control any psychological factors that might impact performance.

## CONCLUSION

Our results demonstrate that acute ischemic preconditioning does not influence the performance of amateur soccer players during the YoYoIE2 test and does not alter the rate of perceived exertion, heart rate (peak and mean) and blood lactate among IPC, SHAM and control. However, as we have not tested the effectiveness of chronic IPC protocols (e.g., for several days or weeks), nor different IPC protocols (e.g., $3 \times 3$-min occlusion/2-min reperfusion) nor other types of YoYo tests (e.g., recovery level I) and mechanistic measurements (e.g., oxygen uptake, reactive oxygen species, muscle blood flow), these could be a future direction.

### Funding
This work was supported by the State Funding Agency of Minas Gerais, Brazil (FAPEMIG). The funders had no role in study design, data collection and analysis, decision to publish, or preparation of the manuscript.

### Grant Disclosures
The following grant information was disclosed by the authors:
State Funding Agency of Minas Gerais, Brazil (FAPEMIG).

### Competing Interests
The authors declare there are no competing interests.

### Author Contributions
- Isabela Coelho Marocolo and André Monteiro Londe performed the experiments, contributed reagents/materials/analysis tools.
- Gustavo Ribeiro da Mota conceived and designed the experiments, analyzed the data, contributed reagents/materials/analysis tools, wrote the paper, prepared figures and/or tables, reviewed drafts of the paper.
- Stephen D. Patterson wrote the paper, reviewed drafts of the paper.
- Octávio Barbosa Neto analyzed the data, contributed reagents/materials/analysis tools, revision of the manuscript.
- Moacir Marocolo conceived and designed the experiments, performed the experiments, analyzed the data, contributed reagents/materials/analysis tools, wrote the paper, prepared figures and/or tables, reviewed drafts of the paper.

### Human Ethics
The following information was supplied relating to ethical approvals (i.e., approving body and any reference numbers):

This study was approved by the local institutional Ethical Committee for Human Experiments (Federal University of Triangulo Mineiro - 993.636/2015) and was performed in accordance with ethical standards in sports science research (Harriss & Atkinson 2015).

## Data Availability

The raw data has been provided as Data S1.

## Supplemental Information

Supplemental information for this article can be found online at http://dx.doi.org/10.7717/peerj.4118#supplemental-information.

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
