# Peer review of "Acute ischemic preconditioning does not influence high-intensity intermittent exercise performance"

_PeerJ, doi:10.7717/peerj.4118_

## Round 0.1 · original submission · Major Revisions

The paper has interesting points but needs substantial improvement before consideration for publication. It is essential to address all reviewers' concerns point-by-point.

Reviewer 1 ·

Basic reporting

Ischemic preconditioning (IPC) has been suggested as a novel treatment that has the potential to improve exercise performance. The current study is aimed to examine the acute effect of IPC on intense intermittent exercise performance and associated physiological markers in amateur soccer players. Subjects were divided into three groups receiving: a) IPC (high pressure exerted on each thigh during blood occlusion phase); b) IPC SHAM (low pressure exerted on each thigh during occlusion phase); and c) control (no treatment). Following the treatment, YoYo intermittent Endurance Test level 2 (YoYoIE2) was performed to evaluate intermittent maximal exercise performance in each group. The results showed that IPC has no significant effect on players’ performance of YoYoIE2. In addition, there was no marked difference in lactate concentration, rate of perceived exertion, and heart rate between the three groups. These results show that IPC does not have any influence on vigorous intermittent exercise duration in amateur soccer players or related biomarkers.

Experimental design

see below

Validity of the findings

see below

Additional comments

Points:
1. The following sentence is shown in lines 138-139, “Mean HR and peak HR presented no difference (p=0.17) among the three trials (Table 1).” However, in table 1, the p value of HR peak is presented as 0.06 instead of 0.17. Could the authors explain which the correct p value for HR peak is?

2. It is concluded in this article that “IPC does not influence high-intensity intermittent exercise performance in amateur soccer players”. However, the study only tested IPC protocol consisting of four cycles of 5-min occlusion. Could the authors justify why they use this IPC protocol? Both the number of cycles and the duration of occlusion may have significant effects on exercise performance. To confirm their conclusion, different IPC protocols (such as increasing or decreasing the IPC cycles) should be tested for their effects on exercise performance. Furthermore, the current study should clarify in their conclusion that only acute IPC did not have effect on their exercise performance, because the study has not tested the effectiveness of chronic IPC protocols (for several days or weeks), which could be a future direction.

3. “HR” shown in line 29 in abstract should be defined first time shown in text; IPC in figure 2 should be defined in its figure legend.

4. The authors only measured 13 players, whose sample size is too small to justify their conclusion. Why did the authors say this suffices for statistical points? Please discuss the limitation of this with convincing details.

5. The discussion on the molecular levels of such study is not in-depth. Reactive oxygen species play an important role in both IPC and exercise. The authors need to discuss this aspect by adding a few sentences like this:
“Exercise can strengthen the body against ROS attack through redox-associated IPC preconditioning. This exercise-induced adaptation involves the enhancement of cellular antioxidant capacity to reduce ROS levels. Increased ROS generation in striated muscles plays a critical role in exercise adaptation as it can trigger exercise-mediated adaptive responses through redox regulations…..” Here for above the authors should cite the following recent key papers by experts in the fields: 1) Front Physiol. 2016; 7: 486. 2016. doi: 10.3389/fphys.2016.00486; 2) Cell death & disease 4 (9), e787, 2013.

6. In lines 41 and 42, change “its supposed exercise enhancement is still to be clarified” to “its supposed exercise enhancement has yet to be clarified”

7. In lines 42-44, change the sentence “Among potential mechanisms of IPC to exercise enhancement, are highlighted…” to “Potential mechanisms of IPC exercise enhancement are hyperemia (5-6 fold increase in muscle blood flow) during reperfusion (Libonati et al. 2001), attenuated ATP depletion, and increased phosphocreative production/oxygen uptake during the reperfusion phase (Andreas et al. 2011).”

8. In lines 51-53, change the sentence “In this context, the YoYo intermittent endurance test…” to “The YoYo intermittent endurance test level 2 (YoYoIE2) is useful to simulate high-intensity intermittent exercise similar to the activity seen in team sports, and is effective due to its simplicity, reproducibility, and its association with both V ́ O2 and citrate synthase activity in untrained men”

9. Please clarify what team sports are being simulated by the YoYo test. It is too vague to attribute high-intensity intermittent exercise to team sports in general as every team sport is very different from one another.

10. In lines 55-59, it is stated that “aerobic energy production plays a more important role in fatigue resistance in the untrained during the YoYoIE2 compared with trained soccer players” however, high-intensity intermittent exercise implies predominantly anaerobic energy production. What is the role of aerobic energy production in the YoYoIE2 test?

11. In lines 170-171, change “These same authors have been argued that there are IPC responders and non-responders” to “These same authors have argued that there are IPC responders and non-responders”. Also, please clarify what is meant by IPC responders and non-responders and what differs between the two.

Reviewer 2 ·

Basic reporting

The study investigated the acute effects of ischemia preconditioning on hign intensity intermittent exercise performance in amateur soccer players. The authors found out that IPS had no influence on the exercise performance.

Experimental design

It's good.

Validity of the findings

OK.

Additional comments

Please confirm the IPS protocol did have a effective stimulation on the subjects. Any marker from the blood immediately can be used to evaluate the response of subjects to IPS before the exercise?

Reviewer 3 ·

Basic reporting

The purpose of this study is to exam the acute effect of ischemic preconditioning (IPC) on a high-intensity intermittent exercise performance and physiological indicators in amateur soccer players. Paper is poorly written and lack of clarity in some sentences (please see below for the details). In addition, there are a lack of details in methodology (for example, what is the sampling time for lactate?). Lastly, there is a lack of depth in discussion.

Experimental design

According to the previous studies (Patterson et al. 2015 & Incognito et al. 2016), IPC does demonstrate that there is an improved peak and mean power output during the early stage of repeated sprint event as well increase the maximal oxygen consumption during incremental bicycle exercise. My understanding is that the time interval between IPC and exercise testing should be essential in the impact of the IPC. However, there is a lack of information of the time interval between IPC and Yo-Yo intermittent endurance test.

Although this study included three groups (IPC, SHAM, Control) with single-blinded design, it is still very different to rule out the psychological effect of the participant. The significant finding form the previous study could be partially explained by the highly motivated participant following IPC.

Since there was IPC non-responder and responder individual difference in soccer players, having a larger sample size might make more sense.

Incorporating EKG, ventilation, R , VO2, peak power, and maximal oxygen measurements in this study will strengthen the quality of this paper.

Validity of the findings

no comment

Additional comments

Introduction
Line 51-59: there is a lack of clarity in these sentences. For example “ being associated with both VO2 and citrate synthase activity in untrained men” I am not sure what this means? Based on the logic flow, author assumed that untrained is similar to amateur which I am not completely agree with.

Discussion:
There is a lack of discussion in depth in physiological explanation for the absence of significant finding during YoYoIE2 test following IPC in comparison to other studies, although author briefly mentioned that this difference is probably due to the type of exercise. It seems like that IPC had a certain acute effect in exercise performance from previous study, however how long this acute effect could last was not clear. The nature of the intermittent maximal test might counterbalance the acute effect following IPC.

---

## Round 0.2 · accepted · Accept

Your final revision have been accepted. Thank you.

Reviewer 1 ·

Basic reporting

The authors have addressed my major concerns.

Experimental design

See below

Validity of the findings

See below

Additional comments

The authors have addressed my major concerns. I accept it in the latest version.

Reviewer 2 ·

Basic reporting

The study investigated the acute effects of ischemia preconditioning on high intensity intermittent exercise performance in amateur soccer players. The authors found out that IPS had no influence on the exercise performance.

Experimental design

The ischemic preconditioning protocol is under the steady state where muscle was not working and in the rest condition. Please consider to include some level of exercise in the ischemic preconditioning protocol that may generate better preconditioning effects.

Validity of the findings

OK

Additional comments

NO